# Functional Analysis of the Plasma Membrane H^+^-ATPases of *Ustilago maydis*

**DOI:** 10.3390/jof8060550

**Published:** 2022-05-24

**Authors:** Melissa Vázquez-Carrada, Michael Feldbrügge, Dario Rafael Olicón-Hernández, Guadalupe Guerra-Sánchez, Juan Pablo Pardo

**Affiliations:** 1Departamento de Microbiología, Escuela Nacional de Ciencias Biológicas, Instituto Politécnico Nacional, Prolongación de Carpio y Plan de Ayala S/N Santo Tomás, Ciudad de Mexico C.P. 11340, Mexico; melivqzc@gmail.com (M.V.-C.); doliconh@ipn.mx (D.R.O.-H.); 2Institute for Microbiology, Cluster of Excellence on Plant Sciences, Heinrich Heine University Düsseldorf, 40204 Düsseldorf, Germany; feldbrue@hhu.de; 3Departamento de Bioquímica, Facultad de Medicina, Universidad Nacional Autónoma de Mexico, Circuito Interior S/N, Ciudad Universitaria, Ciudad de Mexico C.P. 04510, Mexico

**Keywords:** H^+^-ATPases, P-type ATPases, plasma membrane, proton pump ATPase, *Ustilago maydis*

## Abstract

Plasma membrane H^+^-ATPases of fungi, yeasts, and plants act as proton pumps to generate an electrochemical gradient, which is essential for secondary transport and intracellular pH maintenance. *Saccharomyces cerevisiae* has two genes (PMA1 and PMA2) encoding H^+^-ATPases. In contrast, plants have a larger number of genes for H^+^-ATPases. In *Ustilago maydis*, a biotrophic basidiomycete that infects corn and teosinte, the presence of two H^+^-ATPase-encoding genes has been described, one with high identity to the fungal enzymes (*pma1,* *UMAG_02851*), and the other similar to the plant H^+^-ATPases (*pma2,* *UMAG_01205*). Unlike *S. cerevisiae*, these two genes are expressed jointly in *U. maydis* sporidia. In the present work, mutants lacking one of these genes (Δ*pma1* and Δ*pma2*) were used to characterize the role of each one of these enzymes in *U. maydis* physiology and to obtain some of their kinetic parameters. To approach this goal, classical biochemical assays were performed. The absence of any of these H^+^-ATPases did not affect the growth or fungal basal metabolism. Membrane potential tests showed that the activity of a single H^+^-ATPase was enough to maintain the proton-motive force. Our results indicated that in *U. maydis*, both H^+^-ATPases work jointly in the generation of the electrochemical proton gradient, which is important for secondary transport of metabolites and regulation of intracellular pH.

## 1. Introduction

Plasma membrane proton ATPases (H^+^-ATPase) catalyze the extrusion of H^+^ from the cell [1,2,3]. Fungal and plant cells contain at least one isoform of H^+^-ATPase that belongs to the P2-type ATPase family, which includes the animal Ca^2+^ and Na^+^/K^+^ pumps [4]. The proton electrochemical gradient generated by the plasma membrane H^+^-ATPases is required for the activity of several uniporters, symporters, and antiporters [5]. Moreover, these enzymes have important roles in intracellular pH regulation [6], growth processes [7,8], and salt-stress tolerance [9]. Furthermore, some studies revealed the importance of plasma membrane H^+^-ATPases during infection and pathogenesis [10].

Under normal conditions, the main *Saccharomyces cerevisiae* plasma membrane H^+^-ATPase (Pma1p) consumes up to 20% of cellular ATP [11]. Given the importance of this master enzyme, it is not surprising that mutations with drastic reductions in its activity are lethal for *S. cerevisiae* [12]. Although this microorganism contains a second plasma membrane H^+^-ATPase gene (*pma*2), this is not essential due to its low expression [13]. Several kinetic studies have been reported for the plasma membrane proton pumps of *S. cerevisiae* [14], *Neurospora crassa* [15,16], and *Schizosaccharomyces pombe* [17]. In contrast with this large amount of information on these three enzymes, there are few studies on the kinetics and regulatory properties of other fungal H^+^-ATPases.

Some fungi, such as *Uromyces fabae* [10] or *Blumeria graminis* [18] contain only a single H^+^-ATPase gene in their genomes. However, similar to *S. cerevisiae*, *U. maydis* contains two plasma membrane H^+^-ATPases [19]. From an evolutionary viewpoint, it is interesting that one of the *U. maydis* H^+^-ATPase is closely related to the fungal ATPases (52% amino acid identity with the Pma1p of *S. cerevisiae*), and the other shows a higher identity with the H^+^-ATPases of plants (48% amino acid identity with the HA11 of *Arabidopsis thaliana*) [19]. In a previous report from this laboratory, we described that both genes are transcribed when yeast-like cells were cultured in YPD and minimal medium (MM) with glucose or ethanol as carbon sources [19]. Moreover, we confirmed the presence of both enzymes at the plasma membrane by mass spectrometry [19]. Kinetic characterization of the ATPase activity in isolated plasma membranes showed Michaelis−Menten kinetics with a Km for ATP of 2.0 mM and a wide pH optimum centered at pH 7 [19]. As both proton pumps are expressed approximately in the same amount, the two enzymes probably contribute to the measured ATPase activity. To solve this problem, we decided to generate two *U. maydis* strains containing only one plasma membrane H^+^-ATPase, ∆*pma*1 and ∆*pma*2. Our main goal in this work was to study the effect of these deletions on some physiological processes, such as growth, acidification of the external media, generation of a membrane potential, ATPase activity at the plasma membrane, and response to some types of stress.

## 2. Materials and Methods

### 2.1. Strains, Culture Media, and Growth Conditions

*U. maydis* strains (Table 1) were maintained in 25% glycerol (*v/v*) at −70 °C. A small aliquot of the frozen cells was transferred to 100 mL of YPD liquid media (5 g yeast extract, 2.5 g peptone, and 10 g dextrose per liter) and incubated for 24 h at 28 °C and 180 rpm. In addition, these cells were used as inoculum for subsequent experiments. For experiments in MM, cells were washed 3 times in sterile distilled water, recovered by centrifugation (1500× *g*, 5 min), and poured into MM containing 1% glucose, 0.3% ammonium nitrate, and 65.5 mL/L of salt solution [20] made of 16 g K_2_HPO_4_, 4 g Na_2_SO_4_, 8 g KCl, 2 g MgSO_4_, 1 g CaCl_2_, and 8 mL trace elements (60 mg de H_3_BO_3_, 140 mg de MnCl_2_·4(H_2_O), 400 mg de ZnCl_2_, 40 mg de Na_2_MoO_4_·2(H_2_O), 100 mg de FeCl_3_·6(H_2_O), and 400 mg de CuSO_4_·5(H_2_O)).

### 2.2. Mutant Sporidia Generation

The *pma1* (*UMAG_02581*) and *pma2* (*UMAG_01205*) genes were deleted with a hygromycin resistance cassette by homologous recombination using the Golden gate *BsaI* cloning system established by Terfrüchte et al. [21]. Strain FB2 was used in our experiments and the in silico analysis was performed on the *Ustilago maydis* genome database *Ensembl fungi* (https://fungi.ensembl.org/Ustilago_maydis/Info/Index, accessed on 4 November 2021) using the Clone Manager Professional, 9.2 software (©Scientific & Educational Software). Flanking regions of the genes of interest were synthesized by PCR (about 1 Kbp) with cleavage sites for the restriction enzymes *BsaI* and *SspI.* To achieve the transforming plasmids, the hygromycin resistance cassette and the PCR-generated flanks were mixed in a one-pot reaction. Subsequently, *E. coli* Top10 was transformed with the plasmids, the plasmid DNA was purified, digested with *SspI*, and protoplasts of the wild-type sporidia were transformed with the linearized DNA using heparin and PEG [22]. The mutant strains were confirmed by PCR and Southern blot. All of the primers and plasmids used in this work are shown in Table 2.

### 2.3. Growth Kinetics and Glucose Consumption

Twenty-five milliliters of YPD were inoculated with wild-type (FB2) or mutant strains (FB2ΔPma1 and FB2ΔPma2) at an initial optical density of 0.04, and cell growth was followed for 72 h by changes in optical density at 600 nm. Aliquots of 100 μL were taken at 0, 24, 48, and 72 h to determine glucose consumption using Glucose-TR kit (Spin-React, Sant Esteve de Bas, Spain). This is based on the spectrophotometric determination of hydrogen peroxide, which is produced during the reaction catalyzed by glucose oxidase.

### 2.4. Oxygen Consumption

Sporidia were harvested by centrifugation at 24, 48, and 72 h of culture, washed three times with cold distilled water, and collected by centrifugation (1500× *g*). The pellet was resuspended with distilled water in a 1:1 ratio (mL/wet weight) and used for oxygen consumption experiments. The respiratory activity was measured with a Clark-type electrode in a final volume of 1.5 mL [23]. The experiments were performed at 30 °C with 10–20 µL of sporidia in an air-saturated buffer containing Tris-HCl 10 mM pH 7.5 and 10 mM glucose. The oxygen consumption rate was calculated from the initial slope of the oxygen versus time trace and expressed as µmol O_2_·(min·mg dry weight)^−1^.

### 2.5. H^+^-ATPase Activity Indirect Assay

The optical density of sporidia cultures was adjusted to 0.05 and then the suspension was diluted at 10-, 100-, and 1000-fold. In addition, 3 µL-aliquots were poured on plates containing: (a) MM-agar (10 g glucose, 3 g NH_4_NO_3_, 65.5 mL salt solution, and 20 g agar per liter [20] adjusted to different pH values: 5.5, 6.5 or 7.5) supplemented with 0.6 M CH_3_COOH; or (b) modified synthetic dextrose minimal medium agar (SDM) (10 g glucose, 7 g yeast nitrogen base (YNB), 50 mM succinic acid pH 5.5, 20 g agar per liter) supplemented with 1.2 M NaCl, 0.12 KCl [24], and/or 30–50 mM nourseothricin (NTC) (Sigma-Aldrich, Steinheim, Germany). Plates were incubated at 28 °C for 48 h and growth was evaluated.

### 2.6. Effect of Different Stress Conditions under Cell Growth

Cells were treated as in the previous section and poured on plates containing SDM supplemented with 1.8 M sorbitol, 1.0 M glycerol, 1.2 M NaCl, or 0.12 M KCl, and incubated during 48 h at 28 °C, except for cold stress, where plates were cultured at 5 °C.

### 2.7. Isolation of Total Plasma Membrane Fraction

Sporidia were grown in a YPD medium for 24 h at 28 °C, harvested by centrifugation, washed two times with 50 mM Tris pH 7.5, and resuspended in the same Tris buffer. The plasma membrane was obtained in accordance with the protocol described by Serrano [25]. Sporidia were incubated with 5 mM β-mercaptoethanol for 10 min at 28 °C. Next, sporidia were washed twice with a solution containing 600 mM (NH_4_)_2_SO_4_ and treated with the *Trichoderma harzianum* lysing enzymes for 60 min at 28 °C. Spheroplasts were collected by centrifugation (1000× *g*, 5 min) and resuspended in a cold solution containing 330 mM sucrose, 50 mM KH_2_PO_4_ pH 7.4, 1 mM EDTA, and 1 mM of phenylmethylsulfonyl fluoride (PMSF). Thereafter, all of the procedures were carried out at 4 °C. Spheroplasts were lysed with 30 strokes in a Potter-Elvehjem glass homogenizer. The suspension was centrifuged for 5 min at 150× *g* and the supernatant was centrifuged for 10 min at 28,000× *g*. In addition, the new pellet was resuspended in 20% glycerol, 10 mM Tris-HCl, 0.2 mM EDTA, 0.2 mM DTT, pH 7.5, and homogenized, as previously, only 10 times. The resultant solution was poured on top of the sucrose gradient: 1 volume of sample; 1 volume of 53.5% sucrose, 10 mM Tris-HCl, 0.2 mM EDTA, 0.2 mM DTT, pH 7.5; and 2 volumes of 43.4% sucrose, 10 mM Tris-HCl, 0.2 mM EDTA, 0.2 mM DTT, pH 7.5. The sucrose gradient was centrifuged at 70,000× *g* for 4 h. After this time, the plasma membrane fraction was collected from the interphase of the 53.5/43.5% sucrose gradient and centrifuged at 41,000× *g* for 30 min. The pellet containing the total plasma membrane fraction (TMF) was resuspended in a small volume of 20% glycerol, 10 mM Tris-HCl, 0.2 mM EDTA, 0.2 mM DTT, pH 7.5 in addition to 1 mM fresh PMSF and used immediately for the enzyme activity assay or stored at −70 °C.

### 2.8. Enzymatic Activity and Kinetic Studies

ATPase activity was determined as reported by Nakamoto et al. [26] and Robles-Martínez et al. [19]. TMF was added to a reaction mixture containing 10 mM MES, 5 mM MgCl_2_, 5 mM phosphoenolpyruvate, 5 mM Na_2_ATP, 10 mM NaN_3_, 50 mM KNO_3_, and 5 μL/mL rabbit muscle pyruvate kinase to complete the ATP regenerating system. Tris-base was used to adjust the pH of this mixture to 6.5. ATPase activity was measured in the presence or absence of 100 µM Na_3_VO_4_ and incubated at 30 °C. The reaction was stopped at different times (10, 20, 30, and 40 min) by the addition of trichloroacetic acid (6% final concentration) and the tubes were placed immediately in an ice bath. Inorganic phosphate released from ATP was measured in accordance with to Fiske and Subbarrow [27]. After 30 min at room temperature, a blue complex was formed and the absorbance was measured at 660 nm. ATPase activity was calculated from the slope of the linear relationship between phosphate production and incubation time. Specific activity is reported as nmol Pi released·(min·mg protein)^−1^.

### 2.9. Proton Pumping Rate

Proton pumping rate was measured as previously described by Robles-Martínez et al. [19]. Sporidia were collected as described for the oxygen consumption assay and 10^9^ cells were added to a solution containing 200 mM glucose, 20 mM KCl, and 0.2 mM MES pH 6.5 (KOH). The pH change was recorded with a pH electrode for 5 min and the pumping rate was calculated from the initial slope of the curve and expressed as pH units·(min·10^9^ cells)^−1^.

### 2.10. Intracellular pH

*U. maydis* sporidia were electroporated as reported by Peña et al. [28]. Sporidia suspension was placed in an electroporation cuvette containing 2.8 mM pyranine. In addition, a pulse was applied at 1500 V, 25 µF, and 200 Ω for 3.0 ms with a BioRad Gene Pulser. Next, electroporated sporidia were washed, resuspended in distilled water, and incubated in 10 mM MES buffer (adjusted to pH 6.0 with triethanolamine) in a final volume of 2 mL. Glucose (10 mM), KCl (10 mM KCl), and CCCP (4 μM) were added at the indicated times. Changes in pyranine fluorescence were measured at 460–560 nm as described by Calahorra et al., 1998 [29] in a DMX-1000 spectrofluorometer (SLM Instruments; Urbana, IL, USA) with a cell compartment thermostat at 30 °C and a magnetic stirrer, connected to an acquisition system. Moreover, 100 mM NH_4_OH and 100 mM propionic acid were added to reach the maximum and minimum fluorescence values, respectively, to calculate the pH values, in accordance with the Henderson–Hasselbalch equation [28].

### 2.11. Estimation of Plasma Membrane Electrical Potential

Plasma membrane electrical potential was estimated as described by Peña et al. [30], with some modifications. Fluorescence of the fluorophore 3,30-dipropylthiacarbocyanine (DiSC3(3), cyanine), was used to follow the generation of the membrane potential, using a wavelength of 540 nm for excitation and 590 nm for emission. The reaction mixture contained 25 mg cells, 20 mM MES-TEA buffer pH 7.0, 100 mM glucose, and 250 μM BaCl in a final volume of 2.0 mL. When indicated, 0.25 μM DiSC3(3) and 10 μM CCCP were added to the suspension. Changes in fluorescence (540–590 nm) were followed against time in a DMX-1000 spectrofluorometer at 30 °C.

### 2.12. Determination of Protein

Samples were treated with 0.017% deoxycholate and precipitated with 6.0% trichloroacetic acid [31]. Following centrifugation at 3000× *g* for 30 min at 4 °C, the protein content was determined as described by Lowry et al. [32]. Bovine serum albumin was used as standard.

## 3. Results

### 3.1. Growth and Basal Metabolism

Fungal and plant plasma membrane proton ATPases are central players in the cell physiology [5,8]. The importance of these enzymes can be exemplified with the two genes for plasma membrane H^+^-ATPases in the ascomycete *S. cerevisiae*. While *pma*1 is essential to yeast, deletion of *pma*2 has no effect on cell physiology [12]. As *U. maydis* contains two genes for plasma membrane H^+^-ATPases, it was interesting to study the participation of each enzyme in cell physiology and viability. Therefore, mutants and wild-type strains were cultured in YPD and metabolic capacities were measured. Aliquots were withdrawn from the culture media at the indicated times to determine optical density at 600 nm, glucose consumption, and cellular respiratory activity. In contrast with the Pma1p of *S. cerevisiae*, none of the H^+^-ATPases was essential for cell viability. As shown in Figure 1A, cell growth in the YPD medium was the same in mutant and wild-type strains. There was no statistically significant difference in the duplication time of cells lacking one H^+^-ATPase (Table 3). Our results indicate two important points: First, that none of the H^+^-ATPases is essential for cell viability in *U. maydis*, and second, that mutant cells with only one H^+^-ATPase gene grow in a similar manner to the wild-type strain.

One-way ANOVA analysis was carried out to uncover significant differences among the duplication times of the three strains. No statistically significant difference was found between strains using *p* < 0.05 as a threshold. Data were obtained from three independent experiments (n = 3).

Next, we analyzed the consumption of glucose by cells as a measure of energy metabolism and growth. In agreement with the lack of effect on growth kinetics, the absence of one plasma membrane H^+^-ATPase did not affect glucose consumption by cells (Figure 1B). We found that cells grown in batch mode at 180 rpm entered the stationary phase at around 30–40 h of culture, when the optical density reached a value around 3.4 (Figure 1A). As expected from a fully respiratory microorganism without fermentation capacity, glucose in the culture media at the end of the incubation (72 h) was still high (between 50–60% of the initial glucose concentration) (Figure 1B), suggesting that the decrease in an essential factor, such as the dissolved oxygen, limited cell growth at high cell densities.

Mitochondria play a key role in energy metabolism. Approximately 85–90% of the cellular ATP is synthesized in mitochondria [33] and a large proportion of this ATP is consumed in some organisms by the plasma membrane H^+^-ATPases [5]. Based on the functional connection between mitochondria and plasma membrane, we studied the oxygen consumption activity in the H^+^-ATPase mutants. Figure 2C shows that the respiration rate was similar in all strains, regardless of the culture time. Taken together, the results show that deletion of a single H^+^-ATPase in *U. maydis* did not affect cell growth, glucose consumption, and mitochondrial respiratory activity.

### 3.2. H^+^-ATPase Activity

Deletion of one ATPase gene in *U. maydis* could result in different outcomes. One possibility is that the mutation is deleterious for cell survival, although our previous experiments discarded this option. Another possibility is the overexpression of the remaining H^+^-ATPase to compensate for the decrease in H^+^-ATPase activity at the plasma membrane. Contrary to this hypothesis, the expression of both ATPases might be constant and we should observe a decrease in plasma membrane ATPase activity. To explore the cell response to the deletion of one of the two H^+^-ATPase genes, we isolated the plasma membrane of wild-type and mutant strains and measured the ATPase activity at different concentrations of ATP. For the isolation of a good plasma membrane preparation, we followed the procedure described by Serrano [25] with minor modifications. As shown in Figure 2A, the ATPase activity of this preparation was sensitive to vanadate, an essential characteristic of the P-type ATPases family [34,35,36,37]. Accordingly, the vanadate-sensitive H^+^-ATPase activity of the *U. maydis* plasma membrane was obtained by subtracting the activity in the absence of inhibitors minus the activity in the presence of vanadate (Figure 2A). Figure 2B shows that deletion of one ATPase gene did not affect the maximum ATPase activity (Vmax), pointing to a similar ATPase activity at the plasma membrane and suggesting the presence of a regulatory mechanism that compensates the lack of one ATPase. For all strains, the hydrolysis of ATP was consistent with Michaelis–Menten kinetics (Figure 2B). We found similar Vmax and Km values for the H^+^-ATPase activity in the three strains (Figure 2C,D). Noteworthy is the low specific activity of the plasma membrane ATPase in *U. maydis*. Compared with *S. cerevisiae* (1 μmol·(min·mg)^-1^), the specific activity in *Ustilago* is 25 times smaller (30–40 nmol·(min·mg)^−1^). The results suggest that both H^+^-ATPases are expressed at the plasma membrane and that they are active in sporidia.

### 3.3. Proton Pumping Rate and Internal pH

A decrease in the acidification of the external medium and/or a defect in intracellular pH regulation are two plausible results of minor deficiencies in plasma membrane H^+^-ATPase activity in mutant strains. To test the first assumption, we measured the extracellular acidification by cells suspended in a low buffered solution. In addition, to investigate the proton pumping capacity of cells along the growth curve, we analyzed acidification by cells harvested at 24, 48, and 72 h of growth in YPD media. For wild-type and mutant strains, the highest proton pumping activity was found in cells cultured for 24 h (late exponential phase), and it declined when cells entered the stationary phase (Figure 3). For wild-type cells, the decrease in proton pumping activity with time was a smooth function, while for mutant strains the drop in activity was complete in cells from a 48 h culture. Consequently, a statistically significant difference in the means between wild-type and mutant strains was found at 48 h. The results suggest that H^+^-ATPase activity slows down during the stationary phase of growth in YPD media (Figure 3).

To estimate the intracellular pH, cells were electroporated in the presence of pyranine, washed by centrifugation, resuspended in 10 mM MES buffer pH 6.0, and the intracellular pyranine fluorescence was measured at an emission wavelength of 560 nm [28]. We found similar values for the cytosolic pH in wild-type and mutant strains harvested at 24, 48, and 72 h of culture (Table 4). The results suggest that the presence of only one H^+^-ATPase can manage the changes in internal pH.

### 3.4. Effect of an Acidic Load and Nourseothricin on cell Growth

Our results indicate that the expression of only one ATPase is sufficient to support growth, proton-pumping activity, and intracellular pH regulation, but there is a possibility that mutant strains will have a different response under stress conditions. Therefore, we studied cell growth in the presence of sodium acetate at different pH values (5.5, 6.5, 7.5). The rationale of this experiment is that at low pH (5.5), acetic acid will diffuse across the plasma membrane and it will decrease the intracellular pH. Cell growth will depend on the H^+^-ATPase capacity to pump protons outside of the cell. The results show a similar behavior for wild-type and mutant strains (Figure 4A). There was no growth at pH 5.5 and cells were able to grow at pH 6.5 and 7.5.

Next, we studied the growth of wild-type and mutant strains in media containing NTC, an inhibitor of protein synthesis [38]. Defects in the generation of the membrane potential should result in the resistance of mutants to NTC since the transport of this inhibitor depends on the plasma membrane electrical potential [8]. Depolarization of the membrane by KCl should inhibit the uptake of NTC by cells and allow cell growth in the presence of NTC. The results show growth inhibition by NTC in all strains and recovery of growth capacity in the presence of KCl (Figure 4B). A similar behavior was observed in wild-type and mutant strains (Figure 4B), suggesting that the generation of membrane potential by proton ATPases in the mutant strains was not affected.

The results obtained with nourseothricin suggest that the magnitude of the membrane potential of the mutant strains is similar to the one found in wild-type. As this is an indirect assessment of the capacity of cells to generate a plasma membrane electrical potential, we decided to use a specific fluorescent probe for the qualitative determination of the membrane potential, DiSC3(3) [30]. Upon the addition of DiSC3(3) to the mixture containing the cells, the presence of a membrane potential was associated with an increase in the fluorescence intensity, and there was a large drop in fluorescence when the electrical potential was collapsed by the addition of a protonophore, such as CCCP (Figure 5A). Figure 5B shows that the maximum value of fluorescence intensity is similar for all strains, indicating that the magnitude of the membrane potential in the three strains was similar.

### 3.5. Effect of Osmotic/Cold Stress on Cell Growth

A relationship between the activity of plasma membrane H^+^-ATPase and cold or osmotic stress has been reported. This is the case of the *Aeluropus litoralis* plasma membrane proton pump (40). Salt stress induced an upregulation of the H^+^-ATPase [39]. In contrast, cold stress produced a strong inhibition of the *A. thaliana* proton pump [40]. Therefore, we studied the growth of *U. maydis* wild-type and H^+^-ATPase mutant strains in the presence of 1.8 M NaCl, 1.8 M sorbitol, 1.0 M glycerol, and at 5 °C. Assuming an important role of plasma membrane H^+^-ATPases in cell survival, our underlying hypothesis is that the response of mutant strains with only one of the H^+^-ATPase will display a defective response to osmotic or temperature stresses. However, the results show that wild-type and mutant strains grow well in the presence of osmotic stress given by NaCl, sorbitol or glycerol (Figure 6A). We also found that cells were unable to grow at 5 °C (Figure 6B).

## 4. Discussion

Our main goal in this work was to study the effect of *pma1* and *pma2* deletions on some physiological processes, such as growth, acidification of the external medium, ATPase activity at the plasma membrane, and response to some types of stress. To achieve this objective, we generated mutant strains with only one functional H^+^-ATPase, ∆Pma1 or ∆Pma2. The lack of one plasma membrane H^+^-ATPase did not affect the growth capacity of *U. maydis* mutant strains. Glucose consumption nor mitochondrial respiratory activity was altered in mutant strains, suggesting that the capacity of the single enzyme is sufficient to support cell growth. Moreover, the results suggest that there were no changes in *U. maydis* basal metabolism without one H^+^-ATPase. In contrast with these results, Pma1p is essential for *S. cerevisiae* growth [12].

Given the importance of these enzymes in cell physiology, it was surprising that the elimination of one ATPase gene had no impact on *U. maydis* growth and several aspects of cell physiology. Therefore, we analyzed the ATP hydrolysis kinetics in isolated plasma membranes from wild-type and mutant strains to obtain the Vmax, an indirect measure of the amount of each ATPase in the membrane. The results show similar values of Vmax and Km for the wild-type and both ∆Pma1 and ∆Pma2 mutants, indicating a fairly constant specific activity of H^+^-ATPase at the plasma membrane. Interestingly, the specific activity of the plasma membrane ATPase in *U. maydis* is quite low. In *S. cerevisiae,* the specific activity is around 1 μmol·(min·mg)^−1^, while in *Ustilago,* the activity is 25 times smaller (30–40 nmol· (min·mg)^−1^).

Acidification of the external medium depends on both the activity of the H^+^-ATPase [41] and the efflux of metabolites across the plasma membrane [41,42]. Organic acids, such as lactate, malate, and succinate are among the extracellular metabolites released by *S. cerevisiae* during the acidification of the medium [43,44]. The release of pyruvate by yeast *U. maydis* cells has been reported previously [45]. A direct effect of the decrease in plasma membrane ATPase activity would be a lower acidification capacity. The acidification rate was determined at 24, 48, and 72 h in YPD media. The highest rate was obtained at 24 h for all the strains and decayed with time. However, there was no statistically significant difference between mutant and wild-type strains, pointing to similar ATPase activities in all strains. As a consequence of the comparable plasma membrane H^+^-ATPase activities in wild-type and mutant strains, we did not observe changes in intracellular pH values.

As the proton pumping capacity in mutant strains can be a limiting factor for cell survival under stress conditions, we decided to test different types of stress. In accordance with the weak-acid theory [46], protonated and uncharged acetic acid can diffuse across the plasma membrane. Deprotonation of acetic acid inside the cell will decrease the cytosolic pH, and plasma membrane H^+^-ATPases are responsible for the extrusion of protons. In this context, it has been demonstrated that the yeast H^+^-ATPase activity level is an important factor for acid stress tolerance [8,47,48]. Moreover, *S. cerevisiae* mutant strains lacking Ptk2p, a protein-kinase that activates Pma1p, are not able to grow in media supplemented with 0.12 M CH_3_COONa [24]. Therefore, defects in the proton pump will result in intracellular acidification and growth inhibition. We did not find differences between wild-type and mutant strains growing at pH values of 5.5, 6.5, and 7.5 in the presence of 60 mM acetic acid, suggesting that the H^+^-ATPase capacities are comparable in all strains. Similarly, we did not observe differences in the generation of the plasma membrane electrical potential in wild-type and mutant strains, pointing to similar activities of the H^+^-ATPases at the membrane.

## 5. Conclusions

The results suggest that both plasma membrane H^+^-ATPases, Pma1 and Pma2, are simultaneously expressed in *U. maydis* cells. ATPase activity in wild-type and mutant strains containing only one gene for H^+^-ATPases showed Michaelis–Menten kinetics with a Km for the substrate Mg-ATP around 2.0 mM and a Vmax of 30–40 nmol· (min·mg)^−1^. Interestingly, the results also show that the presence of only one type of H^+^-ATPase at the plasma membrane, Pma1 or Pma2, is sufficient to promote cell growth, acidification of the external media, intracellular pH regulation, and generation of membrane potential. Further work is required to study the role of Pma1 and Pma2 during maize infection and other stress conditions. Due to the low specific activity of these enzymes in the *U. maydis* plasma membrane, the expression of these proteins in a heterologous system, such as *S. cerevisiae* should be ideal to analyze their kinetic and regulatory properties.

## Figures and Tables

**Figure 1 jof-08-00550-f001:**
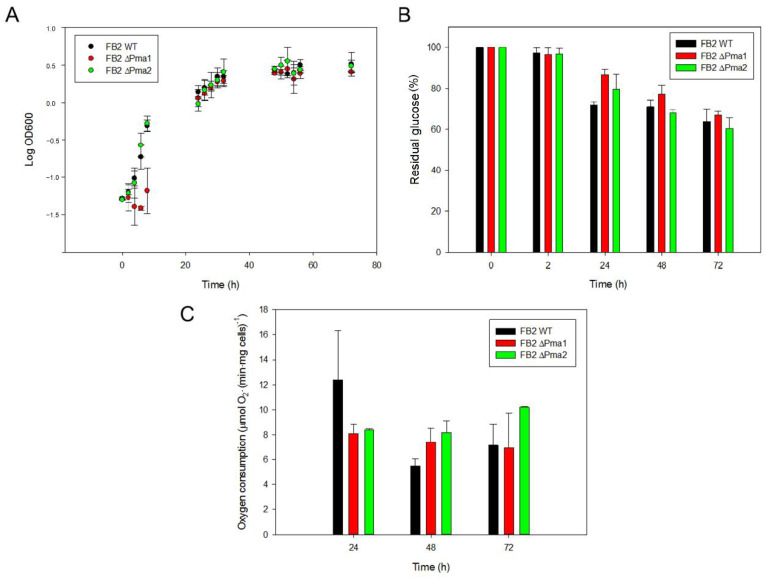
*Ustilago maydis* growth and basal metabolism. *U. maydis* wild-type and mutant cells were grown in 25 mL of YPD media at 28 °C. Aliquots were collected at 0, 2, 24, 48, and 72 h to measure (**A**) optical density at 600 nm, (**B**) the residual glucose concentration in the culture media; the initial concentration of glucose was 55.6 mM, and (**C**) oxygen consumption rate was expressed as µmol O_2_·(min·mg dry weight sporidia)^−1^. One-way ANOVA analysis was carried out to uncover significant differences among the oxygen consumption rates of the three strains. No statistically significant difference was found between strains using *p* < 0.05 as a threshold. Data were obtained from three or more independent experiments (n ≥ 3).

**Figure 2 jof-08-00550-f002:**
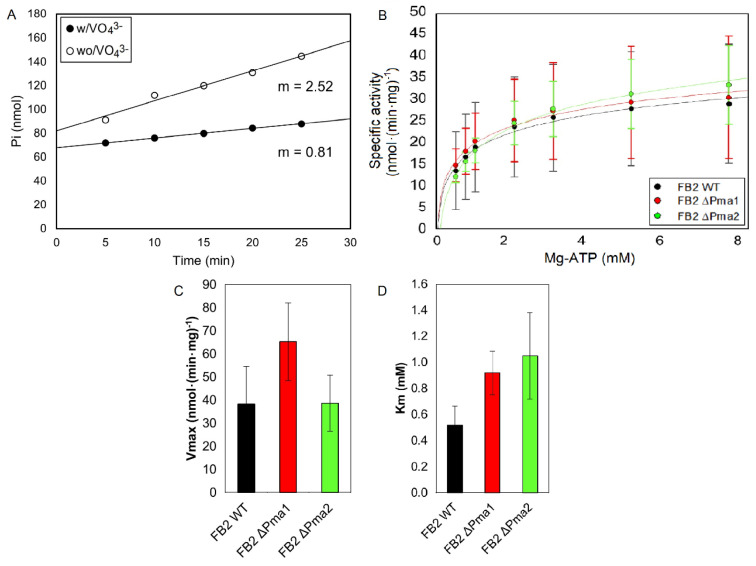
*Ustilago maydis* plasma membrane ATPase activity. *U. maydis* wild-type and mutant strains were grown for 24 h in YPD media. Cells were harvested and the plasma membrane was isolated as described in Materials and Methods. ATPase activity was measured in the presence or absence of orthovanadate, and the vanadate-sensitive ATPase activity was measured at different concentrations of the substrate Mg-ATP. Vmax and Km values were obtained by fitting the data to the Michaelis–Menten equation. Inorganic phosphate released from ATP by the plasma membrane H^+^-ATPase was measured, in accordance with Fiske and Subbarrow [27]. The specific activity is reported as nmol Pi·(min·mg protein)^−1^. (**A**). Time course of phosphate production by the plasma membrane H^+^-ATPase in the presence or absence of 100 μM orthovanadate and 0.5 mM Mg-ATP. (**B**). Variation of ATPase activity with Mg-ATP concentration. (**C**). Maximum velocity values. (**D**). Michaelis–Menten constant values. One-way ANOVA analysis was performed and no statistically significant difference was found between strains using *p* < 0.05 (n ≥ 3). Bars represent the standard error of the mean; m: Slope.

**Figure 3 jof-08-00550-f003:**
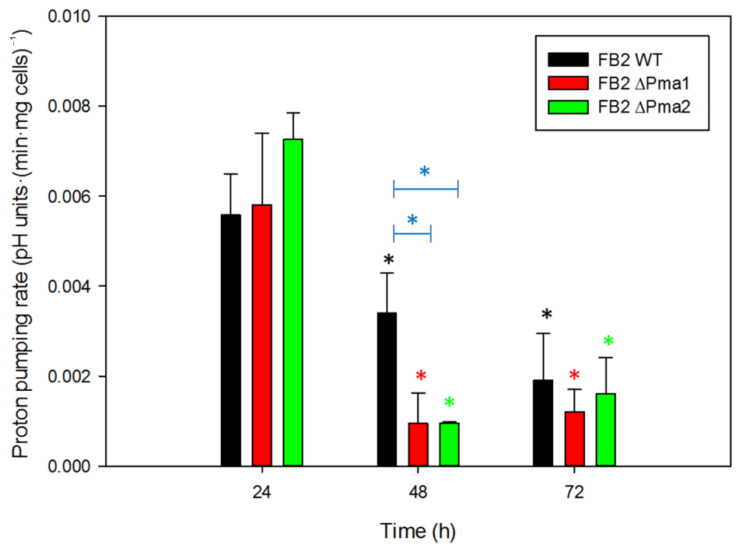
*Ustilago maydis* proton pumping rate. *U. maydis* wild-type and mutant strains were cultured in YPD media, and cells were collected at 24, 48, and 72 h to measure the acidification of the external medium. Proton pumping activity is expressed as pH units·(min·10^9^ cells)^−1^. One-way ANOVA analysis was performed and a statistically significant difference (*p* < 0.05) was found between times (black, red, and green asterisks) and between wild-type and mutant strains at 48 h (blue asterisks) (n ≥ 3). Bars indicate standard deviation.

**Figure 4 jof-08-00550-f004:**
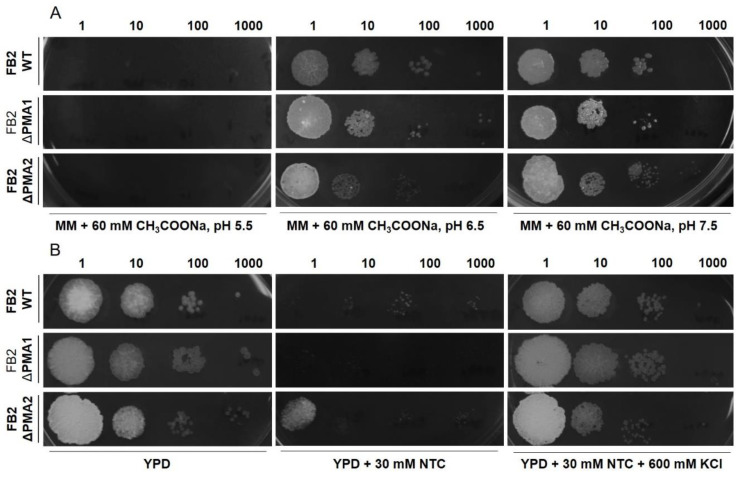
Effect of acetate, nourseothricin, and KCl on cell growth. *U. maydis* wild-type and mutant strains were grown in MM or YPD media to determine the effect of CH_3_COOH (**A**) or NTC (**B**) with or without KCl; NTC: Nourseothricin.

**Figure 5 jof-08-00550-f005:**
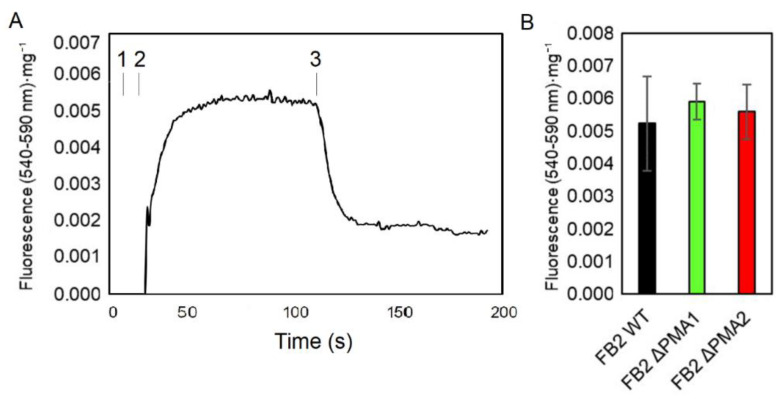
Measurement of membrane potential in *U. maydis* cells. Basidiospores were cultured in YPD for 24 h at 28 °C, harvested, washed, and used to determine the changes in fluorescence of DiSC3(3) against time (**A**). The maximum value of fluorescence was obtained for each strain (**B**). Addition of 1. 0.025 mM DiSC3(3), 2. cells, and 3. 10 μM CCCP at the indicated times. ANOVA analysis was performed and no statistically significant difference was found between strains (*p* < 0.05) (n ≥ 3).

**Figure 6 jof-08-00550-f006:**
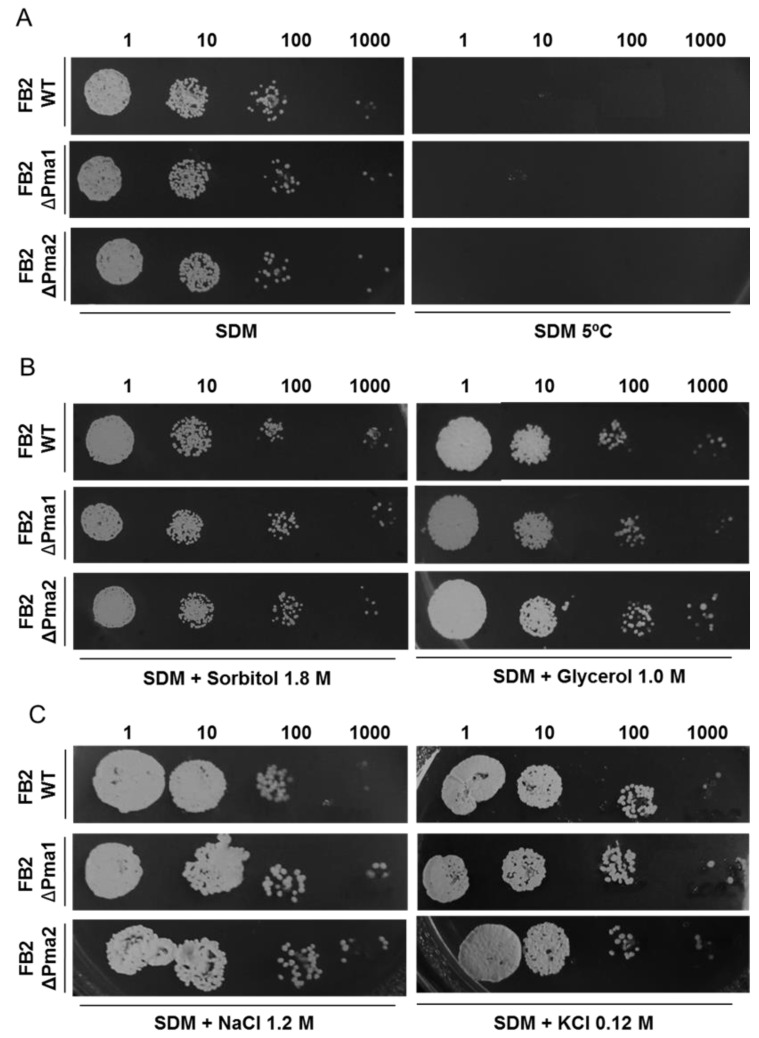
Effect of cold, salt, and osmotic stress on wild-type and mutant strains lacking Pma1 or Pma2. *U. maydis* wild-type and mutant strains were cultured in SDM for 48 h to determine the effect of thermic (5 °C) (**A**), osmotic (sorbitol and glycerol) (**B**), and salt stress (NaCl and KCl) (**C**) on cell growth (n ≥ 3).

**Table 1 jof-08-00550-t001:** *Ustilago maydis* strains used in this work.

Strain	Genotype	Phenotype	Source
FB2 WT	*a2b2; pma1, pma2*	Pma1, Pma2	[19]
FB2 ΔPma1	*a2b2; pma2*	Pma2	This study
FB2 ΔPma2	*a2b2; pma1*	Pma1	This study

**Table 2 jof-08-00550-t002:** Primers and plasmids used in this work.

Primer	Sequence (5′–3′)	Use
ΔPma1		
Upstream flank, Forward primer	CGTAGGCCTCGCTTGTTG	Flank construction
Upstream flank, Reverse primer	GGTCTCGCCTGCAATATTTGTTCTTGCCTCGTCCTGTC	Flank construction
Downstream flank, Forward primer	GGTCTCCAGGCCGATGAAAGAAAAAAGACTACCG	Flank construction
Downstream flank, Reverse primer	GGTCTCCGGCCACCGAGATGCATGCTCACATTC	Flank construction
Diagnostic P1, Forward primer	CGGTGTTGCCATGAACACCGATGGCCAGTG	Diagnostic PCR
Diagnostic P2, Reverse primer	GAGGGCAACGGATTCGAGCTTCTTGGTCTT	Diagnostic PCR
DIG-probe 1, Forward primer	ACGACGTTGTAAAACGACGGCCAG	DIG-probe 1 for Southern blot
DIG-probe 1, Reverse primer	GGTCTCCAGGCCGATGAAAGAAAAAAGACTAC CG	DIG-probe 1 for Southern blot
DIG-probe 2, Forward primer	CGGTGTTGCCATGAACACCGATGGCCAGTG	DIG-probe 2 for Southern blot
DIG-probe 2, Reverse primer	CCAGGTGGAGACAGAGCG	DIG-probe 2 for Southern blot
DIG-probe 3, Forward primer	GGTCTCCGGCCACCGAGATGCATGCTCACATTC	DIG-probe 3 for Southern blot
DIG-probe 3, Reverse primer	TTCACACAGGAAACAGCTATGACC	DIG-probe 3 for Southern blot
ΔPma2		
Upstream flank, Forward primer	GGTCTCGCCTGCAATATTCAACCTCTAAGACTCGCTT	Flank construction
Upstream flank, Reverse primer	GGTCTCCAGGCCTCTGCCTCTTATCTTGCTCTCTTAG	Flank construction
Downstream flank, Forward primer	GGTCTCCGGCCGGGGAAACGTGGAGAAGGTCGCGAAA	Flank construction
Downstream flank, Reverse primer	GGTCTCGCTGCAATATTACCACCCTGTGCCCTCTAG	Flank construction
Diagnostic P1, Forward primer	ACGCTTGACAATCTCGTACTTGTGCTCGGGG	Diagnostic PCR
Diagnostic P2, Reverse primer	GAGGGCAACGGATTCGAGCTTCTTGGTCTT	Diagnostic PCR
DIG-probe 1, Forward primer	GCGCAACTGTTGGGAAGG	DIG-probe 1 for Southern blot
DIG-probe 1, Reverse primer	GGTCTCCAGGCCTCTGCCTCTTATCTTGCTCTCTTAG	DIG-probe 1 for Southern blot
DIG-probe 2, Forward primer	ACGCTTGACAATCTCGTACTTGTGCTCGGGG	DIG-probe 2 for Southern blot
DIG-probe 2, Reverse primer	CCCTCATTGGCTCCGACG	DIG-probe 2 for Southern blot
DIG-probe 3, Forward primer	GGTCTCCGGCCGGGGAAACGTGGAGAAGGTCGCGAAA	DIG-probe 3 for Southern blot
DIG-probe 3, Reverse primer	TGGAAAGCGGGCAGTGAG	DIG-probe 3 for Southern blot
**Plasmid**	**Use**	**Source**
pUMa1810	Transforming plasmid to delete *pma1* gene	This study
pUMa4515	Transforming plasmid to delete *pma2* gene	This study
pUMa1507	Hygromycin resistance cassette	[21]
pUMa1467	Destination vector	[21]

**Table 3 jof-08-00550-t003:** Duplication times of *Ustilago maydis* strains.

Strain	Duplication Time (h)
FB2 WT	2.65 ± 0.90
FB2 ΔPma1	3.52 ± 1.11
FB2 ΔPma2	2.64 ± 0.16

**Table 4 jof-08-00550-t004:** Intracellular pH of *U. maydis* strains.

Strain	Internal pH		
	24 h	48 h	72 h
FB2 WT	7.30 ± 0.05	7.11 ± 0.05	7.66 ± 0.08
FB2 ΔPma1	7.19 ± 0.04	7.39 ± 0.03	7.26 ± 0.63
FB2 ΔPma2	7.07 ± 1.23	7.52 ± 0.22	7.24 ± 0.04

Two-way ANOVA analysis was performed and no statistically significant difference was found between times and strains (*p* < 0.05). (n ≥ 3).

## Data Availability

Not applicable.

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
