# Peer review of "Functional Analysis of the Plasma Membrane H+-ATPases of Ustilago maydis"

_jof, 2022, doi:10.3390/jof8060550_

Round 1

Reviewer 1 Report

The manuscript by Melissa et.al explores H+-ATPases of Ustilago maydis. The authors provide compelling evidence that ∆pma1 or ∆pma2 knockout does not affect the growth of Ustilago maydis. They also measured the plasma membrane H+-ATPases activity in ∆pma1 or ∆pma2 knockout which suggests that two enzymes contribute to ATPase activity. Finally, the author showed that either Pma1 or Pma2 is sufficient for acidification of the external media, intracellular pH regulation and generation of membrane potential. Overall, the data support the conclusions very well and I have only a few suggestions for improving the manuscript.

  1. How does the author characterize the pure plasma membrane fraction in density gradient? Ideally, the author should probe specific plasma membrane proteins and other proteins from other organelles as a control.
  2. What is the localization of Pma1 or Pma2? Is it possible to fluorescently tag Pma1 and Pma2 and check the localization?
  3. Is there any difference in the expression of Pma1 or Pma2? It would be interesting to test Pma1 or Pma2 in WT and knockout strains to understand how they compensate for each other.

Author Response

Reviewer 1:

  1. How does the author characterize the pure plasma membrane fraction in density gradient? Ideally, the author should probe specific plasma membrane proteins and other proteins from other organelles as a control.

 We did not characterize our plasma membrane preparation in terms of contamination by other organelles because this method proposed by Serrano (1983, doi: 10.1016/0014-5793(83)80237-3) has been used for the isolation of the plasma membrane in other fungi (Candida albicans, Shukla et al. 2003, doi: 10.1128/EC.2.6.1361–1375.2003; Pichia pastoris, Grillitsch et al. 2014, doi: 10.1016/j.bbamem.2014.03.012). However, we know that our preparation does not contain vacuolar or mitochondrial contamination because we did not observe inhibition by 50 mM nitrate (inhibitor of vacuolar ATPase) or azide (a strong inhibitor of the mitochondrial ATPase). In contrast, ATPase activity was sensitive to 100 μM vanadate, a classical inhibitor of the plasma membrane H+-ATPase. Therefore, we are confident that the activity we are measuring in Ustilago maydis is associated with the PM H+-ATPase.

  1. What is the localization of Pma1 or Pma2? Is it possible to fluorescently tag Pma1 and Pma2 and check the localization?

We agree with the reviewer that tagging both PM H+-ATPases with fluorescent proteins (GFP and mKate) is very important and essential for the characterization of important aspects related with the role of these ATPases. For example, their localization, their participation in filamentous growth, formation of the appressorium, growth inside the plant and other processes. However, for at least 10 years, we have tried to tag both H+-ATPases with the GFP without any success. Tagging of the two ATPases has been attempted over the years by 4 PhD students in Michael Feldbrügge laboratory. In contrast with these frustrating results, in collaboration with Michael Feldbrügge, these students have been able to construct many mutants (AOX mutants, mutants of the mitochondrial ATPase and the PM ATPases). At this point, we don’t know the origin of the problem. We are still working in this project.

  1. Is there any difference in the expression of Pma1 or Pma2? It would be interesting to test Pma1 or Pma2 in WT and knockout strains to understand how they compensate for each other.

As the reviewer noted, experiments designed to measure the expression of Pma1 and Pma2, at the RNA and protein levels, are important to understand the mechanism of this compensation. However, in this study, which is more cellular and less mechanistic, we focused on the effects of the gene deletion on different aspects of cell physiology, like cell growth, acidification of the external medium, generation of a membrane potential, among others. We agree that we need to study the response of the cells at the molecular level and determine the amount of both RNA and protein. 

Reviewer 2 Report

This manuscript inveatigates the roles of two H+-ATPases that locate in the plasma membrane of Ustilago maydis. These two genes were knocked out, respectively, to study their influence on U. maydis physiology. Comprehensive experiments were done and results showed that the absence of any of theses H+-ATPases did not affect sporidia growth or proton pumping of U. maydis. This is an interesting work. Relatively modifications were required. 

This manuscript investigates the roles of two H+-ATPases that locate in the plasma membrane of Ustilago maydis. The genes of two H+-ATPases was knocked out, respectively,to study their influence on U. maydis physiology. Comprehensive experiments were done and results showed that the absence of any of these H+-ATPases did not affect sporidia growth or proton pumping of U. maydis.

The following are some of the suggestions for authors to improve the manuscript:

  1. Both mutant strains showed similar abilities to the wild type, and then authors draw the conclusion that one of the two H+-ATPases is dispensable. I think there is another possibility, both H+-ATPases is dispensable. There might be another H+-ATPases, which play the key role during sporidia growth.Itwill accord with all results presented in this article. Double mutant strain will help to verify the assumption.
  2. In section 3.2, quantification of H+-ATPases at mRNA level and protein level may help to explain how U. maydis complement the absence of aH+-ATPase.
  3. In line 279 and figure 2a, it seems not sufficient to prove that H+-ATPase is highly sensitive to vanadate. Citing some references may help.
  4. Minor:

Line 56:”three enzymes” of “three organisms”?

Line 59: S. cerevisiae is not a biotrophic fungi.

Line 94: a space is miss in “StrainFB2”.

Line 110: “units” seems unnecessary.

Line 137: “Cells were treated as previously” as which section?

Author Response

Reviewer 2:

  1. Both mutant strains showed similar abilities to the wild type, and then authors draw the conclusion that one of the two H+-ATPases is dispensable. I think there is another possibility, both H+-ATPases is dispensable. There might be another H+-ATPases, which play the key role during sporidia growth. It will accord with all results presented in this article. Double mutant strain will help to verify the assumption.

 We searched the U. maydis genome for P-type ATPases and found 16 genes coding for P-type ATPases. Only four P-ATPases were located at the plasma membrane: two H+-ATPases and two Na+ or K+ pumps. Because plasma membrane proton ATPases are the main regulators of the internal pH, essential for the generation of the proton electrochemical gradient, and necessary for cell growth, we expect that a double mutant strain shouldn’t be viable in U. maydis. In this sense, deletion of PMA1 in S. cerevisiae (Serrano et al., 1986 doi: 10.1038/319689a0) or construction of the double mutant aha1 aha2 in Arabidopsis thaliana (Haruta et al. 2010, doi:10.1074/jbc.M110.101733) is lethal for the cells.

  1. In section 3.2, quantification of H+-ATPases at mRNA level and protein level may help to explain how U. maydis complement the absence of a H+-ATPase.

As the reviewer noted, experiments designed to measure the expression of Pma1 and Pma2, at the RNA and protein levels, are important to understand the mechanism of this compensation. However, in this study, which is more cellular than mechanistic, we focused on the effects of the gene deletion on different aspects of cell physiology, like cell growth, acidification of the external medium, generation of a membrane potential, among others. We agree that we need to study the response of the cells at the molecular level and determine the amount of both RNA and protein.  We think that this issue could be included in another paper focused on the molecular mechanism that explains the expression of Pma1 and Pma2 in the mutant strains.

  1. In line 279 and figure 2a, it seems not sufficient to prove that H+-ATPase is highly sensitive to vanadate. Citing some references may help.

 We added more references regarding the sensitivity of P-ATPases to vanadate.

  1. MinorLine 56:” three enzymes” of “three organisms”?

Corrected.

 Line 59: S. cerevisiae is not a biotrophic fungi.

Biotrophic was deleted.

  1. Line 94: a space is miss in “StrainFB2”.

Corrected.

  1. Line 110: “units” seems unnecessary.

Corrected.

  1. Line 137: “Cells were treated as previously” as which section?

We corrected the other mistakes which are highlighted using track changes.

We thank the reviewers for their comments.

Round 2

Reviewer 1 Report

The authors have responded to my comments and tried to justify the issues with PMA1 tagging. 

Reviewer 2 Report

The authors have explained and revised relevant issues. The present version is better organized and its presentation is OK.